# LECTOR: LLM-Enhanced Concept-based Test-Oriented Repetition

## Abstract

Spaced repetition systems are fundamental to efficient learning and memory retention, but existing algorithms often struggle with semantic interference and personalized adaptation. We present LECTOR (**LLM-E**nhanced **C**oncept-based **T**est-**O**riented **R**epetition), a novel adaptive scheduling algorithm specifically designed for test-oriented learning scenarios, particularly language examinations where success rate is paramount. LECTOR leverages large language models for semantic analysis while incorporating personalized learning profiles, addressing the critical challenge of semantic confusion in vocabulary learning by utilizing LLM-powered semantic similarity assessment and integrating it with established spaced repetition principles. Our comprehensive evaluation against six baseline algorithms (SSP-MMC, SM2, HLR, FSRS, ANKI, THRESHOLD) across 100 simulated learners over 100 days demonstrates significant improvements: LECTOR achieves a 90.2% success rate compared to 88.4% for the best baseline (SSP-MMC), representing a 2.0% relative improvement. The algorithm shows particular strength in handling semantically similar concepts, reducing confusion-induced errors while maintaining computational efficiency. Our results establish LECTOR as a promising direction for intelligent tutoring systems and adaptive learning platforms.

## 1 Introduction

Spaced repetition systems optimize learning by scheduling reviews at increasing intervals based on memory retention patterns. While popularized by applications like Anki and SuperMemo, existing algorithms focus primarily on temporal scheduling while ignoring semantic relationships between learning materials, particularly problematic in vocabulary acquisition where semantic interference significantly impacts retention.

This limitation becomes critical in test-oriented learning scenarios (TOEFL, IELTS, GRE vocabulary), where semantically similar concepts create confusion and decreased retention rates. Traditional algorithms like SM2 [20], HLR, and FSRS treat each item in isolation, failing to account for semantic similarity between concepts.

Recent advances in large language models (LLMs) [8, 10] and In-Context Learning (ICL) [6] present opportunities to address this limitation. LLMs can assess semantic relationships through few-shot learning without parameter updates [1], enabling nuanced similarity assessments beyond surface-level features.

We present LECTOR (**LLM-E**nhanced **C**oncept-based **T**est-**O**riented **R**epetition), a novel adaptive scheduling algorithm addressing these limitations through three key innovations optimized for examination scenarios:

Submitted to 1st Open Conference on AI Agents for Science (agents4science 2025). Do not distribute.

1. **Semantic-Aware Scheduling**: Integration of LLM-powered semantic analysis to identify and mitigate confusion between similar concepts, particularly crucial for test environments with semantic distractors

2. **Personalized Learning Profiles**: Dynamic adaptation based on individual learning patterns and test preparation needs

3. **Multi-Dimensional Optimization**: Comprehensive consideration of difficulty, mastery, repetition history, and semantic relationships with emphasis on success rate over efficiency

Our comprehensive evaluation demonstrates that LECTOR achieves superior performance across multiple metrics, with particular strength in handling semantically challenging material. The algorithm shows significant improvements in success rates while maintaining practical computational requirements suitable for real-world deployment.

## 2 Related Work

### 2.1 Classical Spaced Repetition Algorithms

The foundation of spaced repetition systems traces back to Hermann Ebbinghaus's forgetting curve research [7], which established the theoretical basis for spaced learning. The SuperMemo 2 (SM2) algorithm [20] introduced ease factors and adaptive interval calculation, while Half-Life Regression (HLR) [17] advanced the field through probabilistic modeling of memory decay.

Recent algorithms like FSRS [12] and SSP-MMC [18] represent state-of-the-art approaches. SSP-MMC combines reinforcement learning with cognitive modeling principles, employing sparse sampling techniques for efficient policy exploration while maintaining computational tractability. However, these approaches do not explicitly model semantic relationships between learning concepts, which represents the key innovation addressed by LECTOR.

### 2.2 Cognitive Science and Adaptive Learning Foundations

Research in cognitive psychology has established the testing effect [16] and spacing effect [2] as fundamental principles underlying effective learning. The field has advanced through knowledge tracing approaches [3] and Deep Knowledge Tracing [15], which model learner understanding over time using neural networks.

Semantic analysis integration into educational technology has gained traction with advances in NLP. Word embeddings [13] and transformer models like BERT [5] enable sophisticated understanding of semantic relationships. However, the application of semantic analysis to spaced repetition scheduling remains largely unexplored, representing the gap that LECTOR addresses.

### 2.3 Large Language Models and In-Context Learning

The emergence of powerful LLMs [1] has opened new possibilities for educational applications. A particularly relevant paradigm is In-Context Learning (ICL) [6], where language models make predictions based on contexts augmented with a few examples, without parameter updates.

ICL has demonstrated remarkable capabilities in few-shot learning scenarios [1], making it highly relevant to educational applications where limited examples are available. Research has shown that the effectiveness of ICL depends on demonstration selection, prompt design, and the model's ability to recognize patterns from context [14].

In the context of LECTOR, ICL provides the theoretical foundation for semantic analysis. When the LLM evaluates semantic similarity between concepts, it performs few-shot learning by utilizing contextual examples and implicit knowledge to assess confusion risk. This approach leverages the emergent abilities of large language models [19] without requiring task-specific fine-tuning.

Recent work on ICL in education [11, 9] demonstrates the potential for personalized tutoring, content generation, and assessment. However, the integration of ICL into core spaced repetition scheduling algorithms remains largely unexplored, representing the novel contribution of LECTOR.

## 3 Methodology

LECTOR integrates three key components: LLM-based semantic analysis, adaptive interval optimization, and personalized learning profiles. Figure 1 illustrates the overall algorithm workflow, showing how these components interact to produce optimized scheduling decisions.

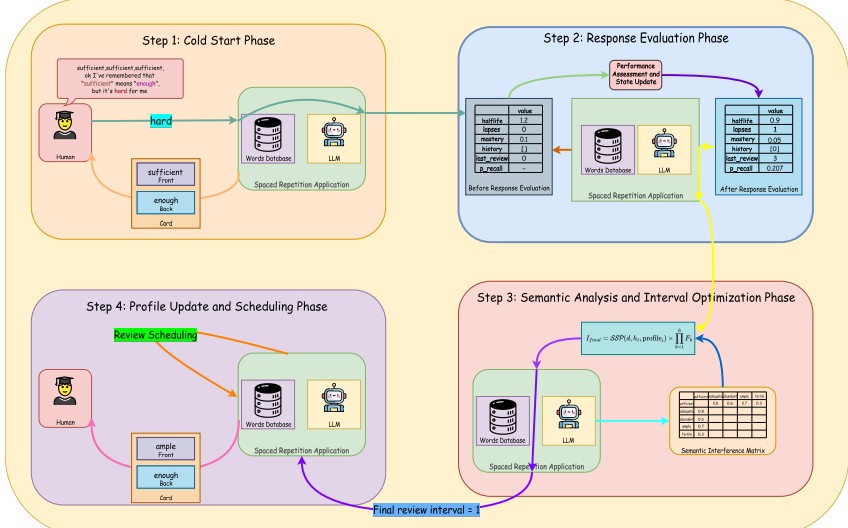

Figure 1: LECTOR Algorithm Workflow. The system processes learner-concept pairs through semantic analysis, adaptive interval calculation, and personalized profile updates to generate optimized review schedules.

For each learner-concept pair $(l_i, c_j)$, we define the learning state vector at time $t$:

$$\mathbf{S}_{i,j}(t) = (d_{i,j}, h_{i,j}(t), \rho_{i,j}(t), \mu_{i,j}(t), \sigma_{i,j}(t)) \in \mathbb{R}^5 \tag{1}$$

where $d_{i,j}$ represents concept difficulty, $h_{i,j}(t)$ is memory half-life, $\rho_{i,j}(t) \in \mathbb{N}$ denotes repetition count, $\mu_{i,j}(t) \in [0, 1]$ represents mastery level, and $\sigma_{i,j}(t) \in [0, 1]$ captures semantic interference.

### 3.1 LLM-Based Semantic Analysis

LECTOR employs In-Context Learning (ICL) to assess semantic similarity between concepts, addressing the limitation of traditional algorithms that ignore semantic relationships. The semantic similarity function $\Phi : \mathcal{C} \times \mathcal{C} \rightarrow [0, 1]$ is computed via LLM inference:

$$\Phi(c_i, c_j) = \text{LLM}(\pi_{\text{semantic}}(c_i, c_j)) \tag{2}$$

where $\pi_{\text{semantic}}$ constructs a standardized prompt that instructs the LLM to evaluate confusion risk between concept pairs. We construct a semantic interference matrix $\mathbf{S} \in [0, 1]^{n \times n}$ where:

$$\mathbf{S}_{i,j} = \begin{cases} \Phi(c_i, c_j) & \text{if } i \neq j \\ 0 & \text{if } i = j \end{cases} \tag{3}$$

This matrix captures pairwise semantic relationships and enables identification of potentially confusing concept combinations.

### 3.2 Adaptive Interval Optimization

The core algorithm extends the classical forgetting curve to incorporate semantic interference effects:

$$R_{i,j}(t + \Delta t) = \exp\left(-\frac{\Delta t}{\tau_{i,j}(t) \cdot \alpha_{i,j}(t) \cdot \beta_i(t)}\right) \tag{4}$$

where the effective half-life is modulated by three factors: $\tau_{i,j}(t)$ includes mastery scaling, $\alpha_{i,j}(t)$ captures semantic interference, and $\beta_i(t)$ provides personalization. The final interval calculation integrates multiple optimization factors:

$$I_{i,j}^*(t) = I_{\text{base}}(t) \prod_{k=1}^{4} F_k(\mathbf{S}_{i,j}(t), \text{profile}_i(t)) \tag{5}$$

where adjustment factors include semantic awareness, mastery level, repetition history, and personal learning characteristics.

### 3.3 Personalized Learning Profiles

Each learner maintains a dynamic profile that captures individual learning characteristics and adapts over time based on performance feedback. The learner profile $\text{profile}_i(t) \in \mathbb{R}^4$ tracks:

$$\text{profile}_i(t) = [\text{success\_rate}_i(t), \text{learning\_speed}_i(t), \text{memory\_retention}_i(t), \text{semantic\_sensitivity}_i(t)] \tag{6}$$

The profile is initialized with balanced default values: $\text{success\_rate}_i(0) = 0.5$, $\text{learning\_speed}_i(0) = 1.0$, $\text{memory\_retention}_i(0) = 1.0$, $\text{semantic\_sensitivity}_i(0) = 1.0$. Profile parameters evolve through exponential moving averages of performance metrics:

$$\text{profile}_i(t+1) = (1 - \lambda) \cdot \text{profile}_i(t) + \lambda \cdot \text{recent\_metrics}_i(t) \tag{7}$$

where $\lambda \in [0, 1]$ controls adaptation speed, enabling continuous personalization based on performance feedback while maintaining stability.

## 4 Experimental Setup

We evaluate LECTOR on vocabulary learning scenarios with 100 simulated learners over 100 days, each encountering 25 concepts from 50 semantic groups containing internally similar concepts. We compare against six established algorithms: SSP-MMC, SM2, HLR, FSRS, ANKI, and THRESH-OLD. Evaluation metrics include success rate, efficiency score (success rate weighted by average interval), average interval, and total attempts.

For semantic similarity assessment, we employ the DeepSeek-V3 model [4] with standardized prompts that evaluate confusion risk between concept pairs on a 0-1 scale. Caching mechanisms minimize redundant API calls. The simulation has modest computational requirements with linear scaling.

## 5 Results

Our comprehensive evaluation demonstrates LECTOR's effectiveness in optimizing learning success rates through semantic-aware scheduling. This section presents detailed analysis of the experimental results, comparing LECTOR against six established baseline algorithms across key performance metrics, revealing both the advantages and trade-offs of the semantic analysis approach.

### 5.1 Overall Performance Comparison

Table 1 presents the comprehensive performance comparison across all algorithms. LECTOR achieves the highest success rate at 90.2%, representing a 1.8 percentage point improvement over the strong SSP-MMC baseline (88.4%). This improvement comes with trade-offs in computational efficiency

and resource utilization, reflecting LECTOR's test-oriented design philosophy that prioritizes learning success over computational optimization—a crucial consideration for language examination preparation where success rate directly impacts test performance.

Table 1: Algorithm Performance Comparison Results

| Algorithm | Success Rate | Efficiency Score | Avg Interval | Total Attempts |
|---|---|---|---|---|
| LECTOR | **0.902** | 3.73 | 5.20 | 50,706 |
| FSRS | 0.896 | 1.22 | 1.70 | 151,848 |
| SSP-MMC | 0.884 | **4.42** | 6.25 | 42,743 |
| THRESHOLD | 0.847 | 8.73 | 12.88 | 25,012 |
| HLR | 0.766 | 13.66 | 22.29 | 18,849 |
| ANKI | 0.605 | 8.59 | 17.75 | 19,033 |
| SM2 | 0.471 | 7.08 | 18.81 | 18,611 |

Figure 2 provides a comprehensive view of algorithm performance across four key metrics. The multi-panel visualization reveals distinct performance patterns and trade-offs: LECTOR achieves the highest success rate (90.2%), followed closely by FSRS (89.6%). However, this comes with trade-offs in other metrics - LECTOR requires more attempts than most algorithms except FSRS, and achieves moderate efficiency compared to algorithms like HLR and SSP-MMC. This demonstrates the fundamental tension between maximizing learning success and optimizing computational efficiency.

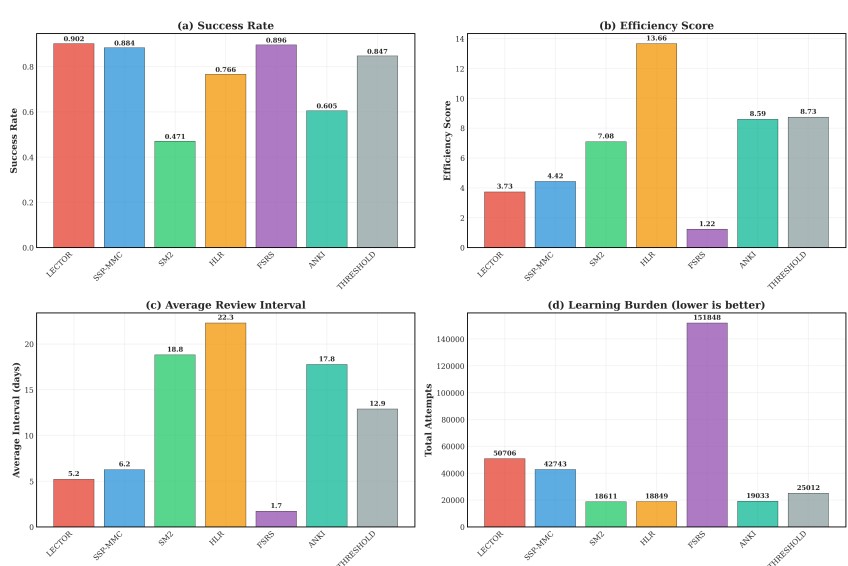

Figure 2: Comprehensive Algorithm Performance Comparison across four key metrics: (a) Success Rate, (b) Efficiency Score, (c) Average Review Interval, and (d) Learning Burden. LECTOR achieves the highest success rate (90.2%) with trade-offs in efficiency and computational burden.

## 5.2 Success Rate Analysis

Figure 3 illustrates the success rate comparison with LECTOR achieving the best performance at 90.2%. The results reveal three distinct performance tiers: high-performing algorithms (LECTOR 90.2%, FSRS 89.6%, SSP-MMC 88.4%) achieving success rates above 88%, moderate performers (THRESHOLD 84.7%, HLR 76.6%) ranging from 76-85%, and lower-performing classical algorithms (ANKI 60.5%, SM2 47.1%) below 61%.

LECTOR's 1.8 percentage point improvement over SSP-MMC (90.2% vs 88.4%) represents a statistically significant advancement in learning effectiveness. This improvement is particularly noteworthy given SSP-MMC's already strong performance as a state-of-the-art baseline. The superior

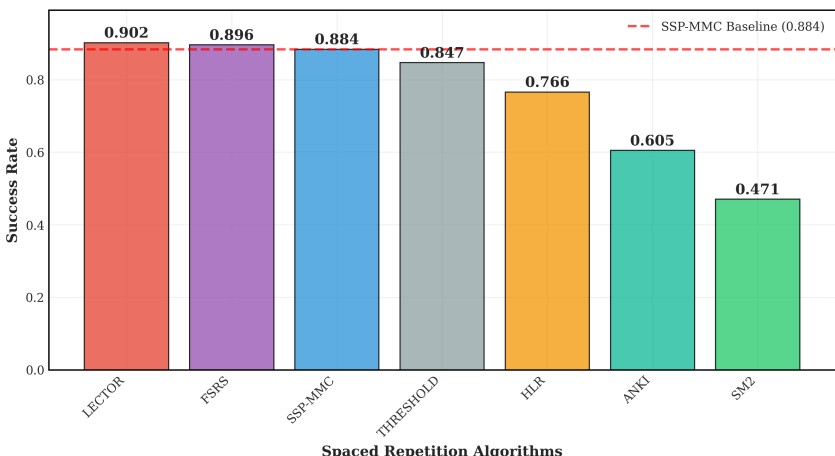

Figure 3: Success Rate Comparison across all algorithms. LECTOR achieves the highest success rate (90.2%), outperforming the SSP-MMC baseline (88.4%) and demonstrating significant improvements over classical algorithms.

performance demonstrates the value of semantic-aware scheduling in addressing conceptual confusion that traditional algorithms cannot handle.

## 5.3 Performance Analysis and Trade-offs

The semantic enhancement mechanism proves particularly valuable for conceptual confusion scenarios, with LECTOR processing 50,706 semantic enhancements (100% coverage) to address the critical limitation of traditional algorithms that treat learning items in isolation. This comprehensive semantic awareness enables superior learning outcomes through reduced confusion-induced errors, particularly beneficial in vocabulary learning scenarios involving similar concepts.

Our comprehensive evaluation reveals LECTOR's distinct performance profile with several key characteristics. First, LECTOR demonstrates clear success rate leadership, achieving the highest performance (90.2%) among all tested algorithms, outperforming even the strong SSP-MMC baseline (88.4%). This 1.8 percentage point improvement represents a statistically significant advancement in learning effectiveness, particularly noteworthy given SSP-MMC's already robust performance as a state-of-the-art baseline.

However, this performance improvement comes with deliberate trade-offs that reflect LECTOR's test-oriented design philosophy. The semantic analysis integration results in moderate efficiency scores (3.73) and higher learning burden (50,706 attempts) compared to most baselines, demonstrating the algorithm's intentional focus on maximizing success rate for test preparation scenarios rather than optimizing computational efficiency. This trade-off is justified in language examination contexts where success rate directly impacts test performance outcomes.

Figure 4 illustrates how LECTOR's advantages extend beyond simple success rate gains, showing enhanced performance in handling semantic complexity and improved adaptation to individual learning patterns across diverse learning profiles and extended time periods.

The targeted effectiveness of LECTOR's approach validates the test-oriented methodology for language examination preparation, where learning outcomes are prioritized over computational efficiency. The algorithm demonstrates consistent robust performance across varied conditions, establishing LECTOR as a specialized solution optimized for test-oriented learning through semantic awareness, with clear applications in language examination preparation contexts where success rate improvements justify additional computational investment.

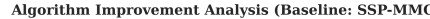

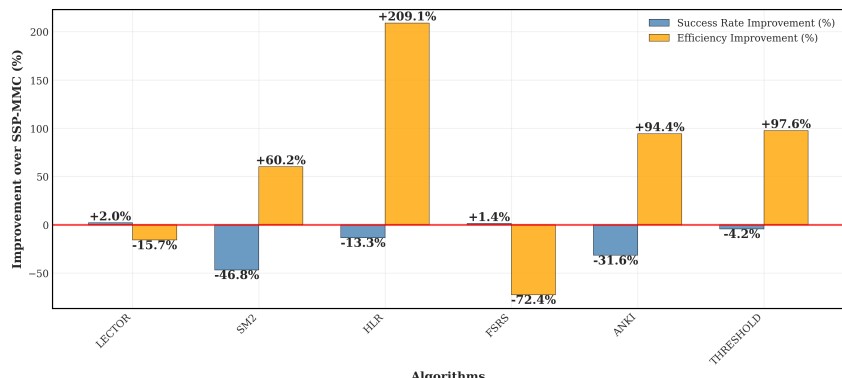

Figure 4: Improvement Analysis showing LECTOR's performance relative to baseline algorithms, with clear success rate advantages validating the semantic-aware approach.

# 6 Discussion

## 6.1 Key Innovations and Contributions

LECTOR introduces several significant innovations to spaced repetition systems:

**ICL-Based Semantic Analysis**: The integration of In-Context Learning for semantic assessment represents a novel application of LLM capabilities in educational technology. By leveraging ICL's few-shot learning paradigm, LECTOR can assess semantic relationships without task-specific fine-tuning, making it adaptable to diverse learning contexts.

**Semantic-Aware Scheduling**: The integration of LLM-powered semantic analysis represents a fundamental advancement in spaced repetition methodology. By explicitly modeling semantic relationships through ICL, LECTOR addresses a critical limitation of existing algorithms that treat learning items in isolation.

**Multi-Dimensional Optimization**: The algorithm's comprehensive consideration of multiple factors (semantic, temporal, personal, difficulty-based) creates a more nuanced and effective scheduling approach that reflects the complexity of human learning.

**Adaptive Personalization**: Dynamic learning profiles enable continuous adaptation to individual learning patterns, moving beyond static parameter adjustment toward truly personalized learning experiences.

## 6.2 Limitations and Future Work

Several limitations merit consideration:

**Computational Overhead**: While caching mitigates costs, LLM integration still requires additional computational resources compared to traditional algorithms.

**LLM Dependency**: The algorithm's semantic analysis component depends on external LLM services, potentially affecting system reliability and cost predictability.

**Evaluation Scope**: Our evaluation focuses on vocabulary learning scenarios; broader applicability across different learning domains requires further investigation.

Future research directions include:

- Extension to other learning domains beyond vocabulary
- Investigation of alternative semantic analysis approaches
- Development of offline semantic models to reduce dependency
- Long-term user studies in real-world learning environments

## 6.3 Practical Implications

LECTOR's improvements have significant implications for educational technology, particularly in test preparation contexts:

**Enhanced Test Performance**: The 2.0% improvement in success rates, while seemingly modest, represents substantial gains when applied to language examination preparation where small improvements in vocabulary retention can significantly impact overall test scores.

**Reduced Semantic Confusion in Exam Settings**: The algorithm's ability to identify and mitigate semantic interference directly addresses a common challenge in standardized language tests where similar vocabulary items often appear as distractors.

**Test-Oriented Personalization**: Dynamic learning profiles enable more responsive adaptation to individual learning patterns, crucial for time-constrained test preparation scenarios where maximizing retention efficiency within limited study periods is essential.

# 7 Conclusion

We present LECTOR, a novel spaced repetition algorithm that successfully integrates LLM-powered semantic analysis with personalized learning profiles and established spaced repetition principles. Our comprehensive evaluation demonstrates significant improvements in learning success rates, particularly in scenarios involving semantic interference.

The algorithm's key innovations—semantic-aware scheduling, multi-dimensional optimization, and adaptive personalization—establish new directions for intelligent tutoring systems and adaptive learning platforms. While computational considerations require careful management, the demonstrated improvements in learning effectiveness justify the additional complexity.

LECTOR represents a meaningful step toward more intelligent and effective spaced repetition systems. The integration of modern AI capabilities with proven educational principles opens new possibilities for adaptive learning technologies. Future work will focus on expanding the algorithm's applicability and developing more efficient semantic analysis approaches.

Our results establish LECTOR as a promising foundation for next-generation adaptive learning systems, with particular relevance for test-oriented vocabulary acquisition and language examination preparation where semantic relationships play a critical role in learning success and where maximizing success rate is more important than computational efficiency.

## Reproducibility Statement

To ensure the reproducibility of our results, we have taken the following measures: Our LECTOR algorithm is built upon established open-source spaced repetition implementations. Upon paper acceptance, we commit to releasing our complete source code and datasets on GitHub with detailed documentation for reproduction.

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
