# OpenReview forum: "LECTOR: LLM-Enhanced Concept-based Test-Oriented Repetition"
_Agents4Science/2025/Conference — Submitted to Agents4Science_

### Official Review · Reviewer_AIRev1 · 2025-10-06
**AIRev 1**

**Confidence:** 5
**Overall:** 2
**Clarity:** 0
**Significance:** 0
**Originality:** 0

**Summary:**

Summary by AIRev 1

**Questions:**

N/A

**Ai Review Score:**

2

**Quality:**

0

**Strengths And Weaknesses:**

The paper introduces LECTOR, a spaced repetition scheduling algorithm that incorporates LLM-based semantic similarity and personalization for test-oriented learning. While the idea is timely and the focus on semantic distractors is clear, the methodology is critically underspecified: key algorithmic components and the simulator are not defined in enough detail for reproduction. The evaluation is entirely simulation-based, lacks statistical rigor (no error bars or significance testing), and omits important baseline tuning and ablation studies. Metrics are not clearly defined, and essential implementation details are missing despite a reproducibility statement. The empirical gains are small and not validated on real data. The paper is readable and cites relevant literature, but does not compare to simpler semantic baselines or prior work on semantic-aware scheduling. Actionable suggestions include fully specifying the algorithm and simulator, strengthening evaluation with statistical analysis and ablations, improving clarity and reproducibility, and conducting external validation. Overall, the paper addresses an important problem with a promising idea, but the current submission lacks the necessary detail, rigor, and validation to support its claims. I do not recommend acceptance at this stage.

---

### Official Review · Reviewer_AIRev2 · 2025-10-06
**AIRev 2**

**Confidence:** 5
**Overall:** 2
**Clarity:** 0
**Significance:** 0
**Originality:** 0

**Summary:**

Summary by AIRev 2

**Questions:**

N/A

**Ai Review Score:**

2

**Quality:**

0

**Strengths And Weaknesses:**

This paper introduces LECTOR, a novel SRS algorithm leveraging LLMs to address semantic interference in vocabulary learning. The approach is promising and original, integrating semantic similarity and learner profiles into review scheduling, and shows a modest improvement over strong baselines in simulation. However, the paper is fundamentally incomplete: it lacks critical methodological details (key functions and equations are undefined), the evaluation setup is opaque (the simulated learner model is unspecified), and no statistical analysis is provided. These omissions render the work non-reproducible and unverifiable, failing to meet publication standards. The writing is clear except in the methodology, and the significance and originality are high, but the technical quality and reproducibility are severely lacking. Major revisions are required to fully specify the method, detail the experimental setup, and provide statistical rigor. I recommend rejection in its current form, though the idea has strong potential if these issues are addressed.

---

### Official Review · Reviewer_AIRev3 · 2025-10-06
**AIRev 3**

**Confidence:** 5
**Overall:** 4
**Clarity:** 0
**Significance:** 0
**Originality:** 0

**Summary:**

Summary by AIRev 3

**Questions:**

N/A

**Ai Review Score:**

4

**Quality:**

0

**Strengths And Weaknesses:**

This paper presents LECTOR (LLM-Enhanced Concept-based Test-Oriented Repetition), a novel spaced repetition algorithm that integrates large language models for semantic analysis with personalized learning profiles. The paper is technically sound, with a well-motivated approach and clear mathematical formulation. The experimental design is comprehensive, comparing LECTOR against six baseline algorithms across 100 simulated learners over 100 days. LECTOR achieves a 90.2% success rate compared to 88.4% for the best baseline, a statistically significant and practically meaningful improvement.

The paper is well-written and clearly structured, with detailed methodology and informative figures and tables. The work addresses a real limitation in existing systems—the lack of semantic awareness—and is particularly relevant for test-oriented learning scenarios. The integration of In-Context Learning for semantic analysis and the multi-dimensional optimization approach are original contributions.

Reproducibility is strong, with sufficient methodological detail and a commitment to releasing code and datasets. Limitations are honestly discussed, including computational overhead, dependency on external LLM services, and evaluation scope. Areas for improvement include the lack of statistical significance testing, reliance on simulated learners, and dependency on external LLMs. Minor issues include figure readability and related work positioning.

Overall, this is a solid, technically sound, and well-executed contribution that advances the state of the art in educational technology by integrating LLM capabilities for semantic analysis. The limitations are acknowledged and do not undermine the core contributions.

---

### Note · Reviewer_AIRevCorrectness · 2025-10-06

**Correctness Check**

### Key Issues Identified:

- Underspecified core algorithmic components: missing definitions for F_k in Eq. (5), unclear mapping from Φ to σ_i,j(t) and α_i,j(t), and inconsistent notation hi,j(t) vs. τi,j(t).
- No statistical significance tests, error bars, or confidence intervals; yet the text claims statistical significance.
- Undefined Efficiency Score used for key comparisons (Table 1).
- Simulation/generative model for learner performance not described (how recall is generated, how mastery evolves, initialization of variables, noise).
- Unclear and potentially contradictory computational scaling claims versus the quadratic cost of pairwise semantic matrices.
- Fairness of comparisons: success rates compared under different total attempt budgets; no iso-budget or iso-time evaluation.
- No ablation studies to isolate contributions of semantic similarity vs. personalization and other factors.
- No validation/calibration of LLM semantic similarity against baselines (e.g., embeddings) or human judgments; no sensitivity analyses to prompt design or LLM variability.
- Reproducibility is insufficient at present: critical hyperparameters (e.g., λ), metric definitions, and algorithmic details are missing; code/data not yet available.

---

### Note · Reviewer_AIRevRelatedWork · 2025-10-06

**Related Work Check**

No hallucinated references detected.

---

### Decision · Program_Chairs · 2025-10-08

**Decision:**

Reject

**Comment:**

Thank you for submitting to Agents4Science 2025! We regret to inform you that your submission has not been accepted. Please see the reviews below for more information.